# *Campylobacter jejuni* in Poultry: Pathogenesis and Control Strategies

**DOI:** 10.3390/microorganisms10112134

**Published:** 2022-10-28

**Authors:** Walid Ghazi Al Hakeem, Shahna Fathima, Revathi Shanmugasundaram, Ramesh K. Selvaraj

**Affiliations:** 1Department of Poultry Science, The University of Georgia, Athens, GA 30602, USA; 2Toxicology and Mycotoxin Research Unit, US National Poultry Research Center, Agricultural Research Service, U.S. Department of Agriculture, Athens, GA 30605, USA

**Keywords:** *Campylobacter jejuni*, broilers, feed additives

## Abstract

*C. jejuni* is the leading cause of human foodborne illness associated with poultry, beef, and pork consumption. *C. jejuni* is highly prevalent in commercial poultry farms, where horizontal transmission from the environment is considered to be the primary source of *C. jejuni*. As an enteric pathogen, *C. jejuni* expresses virulence factors regulated by a two-component system that mediates *C. jejuni*’s ability to survive in the host. *C. jejuni* survives and reproduces in the avian intestinal mucus. The avian intestinal mucus is highly sulfated and sialylated compared with the human mucus modulating *C. jejuni* pathogenicity into a near commensal bacteria in poultry. Birds are usually infected from two to four weeks of age and remain colonized until they reach market age. A small dose of *C. jejuni* (around 35 CFU/mL) is sufficient for successful bird colonization. In the U.S., where chickens are raised under antibiotic-free environments, additional strategies are required to reduce *C. jejuni* prevalence on broilers farms. Strict biosecurity measures can decrease *C. jejuni* prevalence by more than 50% in broilers at market age. Vaccination and probiotics, prebiotics, synbiotics, organic acids, bacteriophages, bacteriocins, and quorum sensing inhibitors supplementation can improve gut health and competitively exclude *C. jejuni* load in broilers. Most of the mentioned strategies showed promising results; however, they are not fully implemented in poultry production. Current knowledge on *C. jejuni*’s morphology, source of transmission, pathogenesis in poultry, and available preharvest strategies to decrease *C. jejuni* colonization in broilers are addressed in this review.

## 1. Introduction

*C. jejuni* was first recognized in 1886 by Escherich as he described the *C. jejuni* as a spiral bacteria isolated from the colon of dead children [1]. Escherich also identified the *C. jejuni* microscopically in stool specimens of children who suffered from diarrhea without being able to culture it on solid agar [1]. In 1909, a *Vibrio*-like bacterium was frequently isolated from aborted fetuses [2], later named *Vibrio fetus* [3]. Similar reports linked *Vibrio*-like organisms to sterility in cows [4], and dysentery in pigs, and later named *Vibrio jejuni* [5].

Similarly, several reports noted the presence of *Vibrio fetus* in the blood of pregnant women [6] and the blood of people associated with outbreaks related to consumption of milk contaminated with *Vibrio fetus* [7]. The absence of a proper isolation method for *Vibrio fetus* (*C. jejuni*) from feces resulted in fewer case reports despite the high prevalence of this pathogen. However, *C. jejuni* was successfully isolated from the stool of a patient suffering from acute enteritis [8]. The development of simpler isolation techniques for culturing *C. jejuni* led to the rapid isolation of this pathogen. In the mid-1980s, *C. jejuni* was recognized as one of the major causes of enterocolitis in humans [9].

*C. jejuni* is the leading cause of human foodborne illness associated with poultry, beef, and pork consumption [10]. *C. jejuni* is found in the gut of warm-blooded animals, with poultry species being the major reservoirs [11]. *C. jejuni* colonizes the ceca of chicken between 2 and 3 weeks of age and reaches around 1 × 10^9^ CFU/g in the ceca at market age [12]. Furthermore, poultry carcass is cross-contaminated at the processing facility due to spillage of intestinal contents. Handling and consuming improperly cooked poultry products account for the majority of *C. jejuni* infections [13]. With the spread of antibiotic resistance across *C. jejuni* isolates, the burden of Campylobacteroisis has increased [14].

The poultry industry is facing several challenges with legislative restrictions on the subtherapeutic use of antibiotics, in addition to the shift in consumers’ preference for “zero” use of antibiotics in poultry production. Therefore, finding an antimicrobial alternative to control *C. jejuni* in poultry production is the need of the hour. Different antibiotic alternatives include prebiotics, probiotics, synbiotic, bacteriocins, bacteriophages, vaccines, and organic acids [15].

This review article focuses on the *C. jejuni*’s morphology, source of transmission, pathogenesis in poultry, and available preharvest strategies to decrease *C. jejuni* colonization in broilers.

## 2. *Campylobacter jejuni* Cellular Structure and Morphology

*C. jejuni* is a gram-negative, corkscrew-shaped, and motile bacteria that belongs to the family of *Campylobacteraceae*. *C. jejuni* is characterized by a spiral/helical morphology with an amphitrichous sheathed flagella responsible for *C. jejuni*’s corkscrew motility [16]. The corkscrew motility or darting motility is a key advantage of *C. jejuni* movement in a highly viscous environment such as the mucus [16]. The enzyme *C. jejuni*’s peptidoglycan peptidase ensures the formation of the helical form, as mutations in peptidoglycan peptidase result in straight body formation [17]. The loss of *C. jejuni*’s helical form results in lower colonization capacity in chickens [17] and lower infectivity in mouse models [17]. *C. jejuni* produces a capsular polysaccharide (CPS) that helps the bacteria evade the immune system and contributes significantly to *C. jejuni* virulence [18]. Intra-strain variation in the capsular polysaccharide results in the formation of 47 serotypes [19]. *C. jejuni* outer membrane is comprised of Lipo-oligosaccharides (LOS) that lack the O-antigen found in lipopolysaccharides (LPS) of many gram-negative bacteria [20]. In some *C. jejuni* strains, the LOS binds with sialic acid resulting in a modified structure that mimics the gangliosides in human neurons [21]. This molecular mimicry plays a central role in developing Guillain–Barré syndrome (GBS) in humans [21].

The presence of phase-variable loci in *C. jejuni* bacteria contributes to its inherent ability to generate different phenotypes and genotypes. These phase-variable loci are mainly located in the CPS, flagella, and LOS, forming new structures to evade the immune system and help *C. jejuni* survive different environmental factors. Different stress conditions lead to morphological changes in *C. jejuni*, for example, oxygen-rich compounds that change from spiral shape to coccoid form [22]. Oxidative stress leads to forming the viable but non-culturable (VPNC) form of *C. jejuni* [23].

## 3. Source and Transmission of *Campylobacter jejuni* in Poultry

*C. jejuni* is a versatile bacterium that occupies different niches and hosts [20]. *C. jejuni* can be found in water and is part of the commensal microbiota of many animals, including poultry [20]. Poultry species are considered the major reservoirs for thermophile *Campylobacter* species, including *C. jejuni*, *C. coli*, and *C. lari*. *C. jejuni* accounts for the majority of campylobacteriosis in humans [20].

*C. jejuni* is highly prevalent in commercial poultry farms, where horizontal transmission from the environment is considered the primary source of *C. jejuni* [24]. Following the infection, broilers rapidly show a high load of *C. jejuni* in the cecal content [25]. Fecal shedding of *C. jejuni* and fecal ingestion is the main source of bird-to-bird transmission in broiler farms [26]. Vertical transmission of *C. jejuni* in broiler farms is controversial as *C. jejuni* is detected in broilers at 2–3 weeks of age, irrespective of *C. jejuni* positive parent flocks [26]. Furthermore, the isolation of *C. jejuni* from eggs in commercial and experimental layer flocks has been unsuccessful [27]. *C. jejuni* led to embryonic mortality in experimentally infected eggs [28], and *C. jejuni* did not survive more than 3–6 h following egg penetration [29,30]. Broiler flocks are usually infected with strains different than the strains detected in the breeder flocks [31], suggesting the negligible role of vertical transmission in *C. jejuni* in broiler flocks. Despite such observations, bacteriological and molecular methods confirmed the [32,33] presence of *C. jejuni* in eggshells. Furthermore, the survival of *C. jejuni* in a viable but not culturable form might be a critical factor behind the unsuccessful isolation of *C. jejuni* from infected eggs, and young hatchlings [34]. Therefore, future studies are needed to elucidate the role of vertical transmission in introducing *C. jejuni* to commercial broiler flocks.

The dry nature of chicken feed and wood shavings decreases the presence of *C. jejuni*, as *C. jejuni*’s viability is hindered by high O_2_ and low moisture levels [35]. Nevertheless, feed and bedding material can be a source of *C. jejuni* as it becomes contaminated by other sources such as fecal material and insects [35]. Reused litter can act as a source of *C. jejuni* infection. However, common litter management practices can limit the spread of *C. jejuni* to the next flock [35]. Unchlorinated water has been suggested as a potential source of *C. jejuni* in broiler farms. Water can act as a vehicle to transmit *C. jejuni* [36] as it requires microaerophilic conditions and cannot grow at a temperature less than 31 °C. *C. jejuni* was found in water lines only after the flock was colonized; however, the strains found in the water lines were not fully present in infected broilers, indicating that water is not the original source of contamination [37,38].

Flies and insects can act as a vector for several pathogens on broiler farms. Flies [39] and beetles [40] may introduce *C. jejuni* into chicken farms from multiple sources such as animal feces and lakes contaminated with *C. jejuni* [41]. The presence of livestock animals on the farm is associated with an increased risk of *C. jejuni* transmission through flies to broiler flocks [41]. *C. jejuni* colonization pattern peaks during summer [42], correlating directly with the insect populations. Therefore, Insects might be an important factor in *C. jejuni* seasonality incidences. Rodents are vectors for pathogens, including *C. jejuni* [43,44]. However, *C. jejuni* strains circulating in humans and livestock differ from those carried by rodents [43]. In some instances, rodents living close to humans and farms may carry the same *C. jejuni* strains [44]. Hence, rodents may not be the original source of *C. jejuni*, yet they remain an important vector that can transmit *C. jejuni* in broiler farms.

*C. jejuni* has been isolated from wild animals. Due to their migratory behaviors, wild animals can spread *C. jejuni* at far distances from the source of infection [45]. The proximity of wild animals in agricultural settings increases the transmission of zoonotic diseases [46]. Furthermore, a wide array of wild animals is hunted for human consumption and can potentially involve in the zoonotic transfer of *C. jejuni* [47]. *C. jejuni* has been isolated from waterfowls [48], songbirds [49], raccoons [50], raptors [51], wild boars [52], and deer [53]. *C. jejuni* isolated from wild animals carries a different lineage from *C. jejuni* isolated from broilers farm [54]. However, *C. jejuni* isolated from wild animals living near broilers farm shows a similar lineage with *C. jejuni* strains found in the broiler farms [54]. The role of wild animals in introducing *C. jejuni* to chicken farms is not fully understood and requires additional studies. Furthermore, farm workers and equipment can have a role in introducing *C. jejuni* to broiler flocks [55]. The movement of contaminated equipment between different farms can potentially transmit *C. jejuni*. Evidence from *C. jejuni* isolation from crates [55], and farmers’ boots [55] proved that contaminated transport crates transmit *C. jejuni* to the slaughterhouse [55]. Reservoirs and routes of transmission of *C. jejuni* are summarized in Figure 1. 

## 4. Pathogenesis of *C. jejuni* in Broilers

*C. jejuni* pathogenesis consists of four main steps: (1) ingestion, (2) acid tolerance and bile resistance, (3) reproduction in mucus, and (4) invasion of epithelial cells [12]. *C. jejuni* infection is transmitted between birds via the fecal–oral route [56]. A small dose of *C. jejuni* (around 35 CFU/mL) is sufficient for successful bird colonization [57]. As an enteric pathogen, *C. jejuni* expresses virulence factors regulated by a two-component system that mediates *C. jejuni*’s ability to survive the gut’s harsh conditions [58]. *Campylobacter* multidrug efflux pump (CmeABC) helps *C. jejuni* in eliminating toxic compounds such as antimicrobials, bile salts, and heavy metals. CmeABC comprises three proteins, a periplasmic protein, an inner membrane protein, and an outer membrane protein [59]. CmeABC gene encodes the multidrug efflux pump in *C. jejuni* and it is regulated through Cme repressor (CmeR) [60]. The presence of bile compounds stimulates the expression of CmeABC, increasing *C. jejuni*’s resistance to bile salts [60]. Mutations in regulator genes related to bile resistance block *C. jejuni*’s colonization ability [61].

*C. jejuni* depends on the two-component system consisting of CheY (cytoplasmic response regulator protein) and CheA (membrane-associated histidine auto kinase sensor) in responding to different chemoattractant/chemorepellents found in different environments [62]. In response to a stimulus, CheA is autophosphorylated, and a phosphate group is transferred to activate CheY. CheY interacts with the flagellar motor switch proteins leading to a clockwise rotation of the flagella [63]. The flagella play a central role in *C. jejuni* motility, adhesion, and invasion of the intestinal epithelial cells [64]. The flagellum consists of seven protofilaments of FlaA and FlaB subunits [64] and is attached to the basal structure through FlgE, which serves as a hook [64]. FlaA is the major Flagellin in *C. jejuni* and is regulated by σ28 promotor [20]. On the other hand, FlaB is the minor flagellin in *C. jejuni* and is regulated by the σ58 promoter [20]. Chemotaxis such as aspartate, glutamate, citrate, and L-fucose upregulates the σ58 gene [65]. FlaA plays a significant role in *C. jejuni*’s initial colonization of the chicken GIT [66]. The FlaA mutant has a ability to decrease the *C. jejuni* colonization in chicken [67]. Furthermore, the flagella include the type III secretion system (T3SS), which is responsible for delivering effector proteins needed for cellular invasion [20]. Thus, mutations in the flagellum lead to a decreased ability in colonization and invasion of intestinal epithelial cells.

The absence of immortalized chicken intestinal cell line hinders the capacity to characterize the mechanism of *C. jejuni* invasion of epithelial cells. In vitro, *C. jejuni* was capable of invading primary avian cells [68,69]. The presence of avian mucus protected the human cell line against the *C. jejuni* invasion [70]. It is well-known that *C. jejuni* survives and reproduces in avian mucus [71]. However, several factors interfere with *C. jejuni*’s capacity to invade the avian epithelial cells in vivo and might explain the near-commensal relationship of *C. jejuni* in avian species. Several differences are observed between humans and avian species in terms of body temperature (37 °C vs. 42 °C), mucus pH (the avian mucus is more acidic), and difference in mucus structure. *C. jejuni* upregulates genes related to metabolism and regulatory systems and downregulates genes related to periplasmic proteins at 37 °C in comparison with 42 °C [72]. This difference in gene expression may explain the *C. jejuni* adaptability and pathogenicity in humans’ intestinal tract. *C. jejuni* upregulates the CadF gene, which is responsible for cell adhesion at 37 °C and 42 °C, indicating the ability of *C. jejuni* to adhere to intestinal cells in humans and avian species [72]. On the other hand, *C. jejuni* isolates showed different gene expressions at 37 °C vs. 42 °C [73]. The difference in gene expression might explain some of the differences in *C. jejuni*’s pathogenicity. However, it might not be enough to justify the complete picture of *C. jejuni* pathogenicity in humans vs. the near-commensal relationship in avian species. It was hypothesized that the pH of the avian mucus confers protection for avian species against *C. jejuni*. However, through in vitro studies, the neutralization of the avian mucus did not diminish its anti-*Campylobacter jejuni* properties [68].

Purified chicken mucin inhibited the adherence and internalization of *C. jejuni* to a human intestinal cell line without affecting *C. jejuni* viability [70]. The oxidation of purified chicken mucin with sodium metaperiodate enabled *C. jejuni* to invade the intestinal cell line [70]. The results indicate the protecting role of o-glycosylated mucin structure in the intestinal cell line against *C. jejuni*.

The avian mucus is highly sulphated and sialylated compared with the human mucus [74]. The comparison between chicken and human mucin structures revealed thirty-three unique structures in chicken mucin [75]. The large intestine in chicken contains the highest sulphated structures, followed by the small intestine and cecum [75]. In chicken, *C. jejuni* colonizes mainly the ceca and, to a lesser extent, in the small and large intestines [76]. Evidence from in vitro studies shows that the purified chicken mucin from the large intestine had a higher inhibition ability against *C. jejuni* compared with the purified chicken mucin from the small intestine and cecum [75]. These results highlighted that presence of sulphated O-glycans is inversely correlated with the concentration of *C. jejuni* in the host. Furthermore, the increased sulfation and sialyation increase the anionic charge in the chicken mucin, creating a charge repulsion effect against *C. jejuni* [74]. These results indicate the role of chicken mucus in modulating *C. jejuni* virulence in avian species. *C. jejuni* pathogenesis is summarized in Figure 2. 

## 5. Control of *C. jejuni* in Broilers: (Preharvest)

*C. jejuni* establishes colonization in the lower intestinal tract of the chicken, particularly in the ceca, within 24 h [65]. *C. jejuni* concentration can reach up to 1 × 10^9^ CFU/g in infected birds [20]. Birds are usually infected from two to four weeks of age and remain colonized until they reach market age [65]. Therefore, control methods are needed to reduce *C. jejuni* prevalence across broiler farms.

### 5.1. Biosecurity

Strict biosecurity measures are the key role in preventing the transmission of *C. jejuni* in broilers’ houses [35]. Identifying the potential sources and methods to detect *C. jejuni* at the farm level are the major steps needed for a successful biosecurity measures. Restricting access to poultry houses is key to maintaining a *C. jejuni*-free flock and following strict biosecurity measures can decrease *C. jejuni* prevalence by more than 50% in broilers at market age [38]. Cleaning and disinfecting poultry houses between cycles can reduce *C. jejuni* prevalence [38]. Furthermore, strict hygiene practices such as boot covers, hand washing, and footbaths can decrease *C. jejuni* transmission [35]. Standard litter management practices are also critical to decrease the presence of *C. jejuni* in the litter, and transmission is reduced with enough downtime between flocks [77].

Partial depopulation (thinning) of broiler flocks can increase the risk of *C. jejuni* transmission on the farm [78]. A strict biosecurity measure during the thinning process is required to ensure a low transmission rate of *C. jejuni* [78]. Furthermore, *C. jejuni* prevalence peaks during summer and early autumn time [79]. This seasonality of *C. jejuni* also correlates with the peak of insect populations [80]. Therefore, strict biosecurity measures are needed during summer, early autumn, and thinning to ensure a low prevalence of *C. jejuni* within the farm.

### 5.2. Probiotics, Prebiotics, and Synbiotic

Probiotics have the ability to improve gut health and prevent enteric diseases in poultry. Several mechanisms of probiotics include (1) antagonism and competitive exclusion of enteric pathogenic bacteria, (2) pH reduction by producing organic acids, (3) bacteriocin production, (4) stimulation/modulation of host immune response, (5) and alteration of virulence factors of enteric pathogens [81]. On the other hand, prebiotics are non-digestible feed ingredients that confer a beneficial effect on the host by promoting the proliferation of beneficial bacteria in the gut [81]. The combination of probiotics and prebiotics is known as synbiotics [81].

The ability of probiotics, prebiotics, and synbiotics to combat *C. jejuni* has been demonstrated in vitro, in vivo, and in field studies. In vitro, the anti-*Campylobacter* activity of probiotics has been carried out in agar-plate diffusion assays [82], co-cultures assays [83], and adhesion and colonization assays using cell lines [82].

Probiotics secrete organic acids that exhibit antimicrobial activity against gram-negative bacteria [84]. In vitro, *E. faecium*, *P. acidilactic*, *L. salivarius*, and *L. reuteri* supernatant inhibited *C. jejuni* growth [84]. Similarly, the supernatant of *L. crispatus* significantly decreased *C. jejuni* growth [85]. *L. crispatus* antimicrobial activity was mediated through the production of organic acids, namely: lactic acid [85]. In ceca, *E. faecalis* strain inoculation decreased *C. jejuni* load to 1 log CFU/g reduction after 6 h post-inoculation [86]. The anti-*Campylobacter* ability of lactic acid-producing bacteria was mediated through the production of organic acids [85].

Probiotic species can disrupt the expression of virulence factors in enteric pathogens [87,88]. The cell-free supernatant media of *L. acidophilus* strain La-5 and *Bifidobacterium longum* strain NCC2705 downregulated ciaB (invasion) and FlaA (motility) [87]. Similarly, the cell-free supernatant of *L. salivarius*, *L. johnsonii*, *L. crispatus*, and *L. gasseri* downregulated flaA, flab, flhA (motility), ciaB (invasion), and AI-2 (quorum sensing molecule autoinducer-2) [89]. These studies demonstrate the ability of probiotics species to attenuate the *C. jejuni* virulence factors. The downregulation of *C. jejuni* motility and invasion genes results in a lower ability to colonize the GIT of broilers and invade human and chicken primary cell lines [89]. *Lactobacillus* spp. Supplementation can modulate the host immune system [89]. *L. salivarius*, *L. johnsonii*, *L. crispatus*, and *L. gasseri* supplementation increased nitric oxide production and phagocytic ability of chicken macrophages, leading to a decrease in the *C. jejuni* load [89]. Furthermore, a mixture of *Lactobacillus* spp. Increased the expression of costimulatory molecules, namely: CD40, CD80, and CD86, in macrophages [89]. The costimulatory molecules are essential to initiate an adaptive and humoral immune response; hence probiotic supplementation can initiate the innate and adaptive immune response against *C. jejuni* [90].

Probiotic bacteria adhere to and occupy gut mucosal surfaces and competitively exclude enteric pathogens. In vitro studies are fundamental to investigating the probiotics’ mechanism of action. On the other hand, in vivo studies provide a comprehensive assessment of probiotics’ ability to benefit the host. Not all promising results in vitro are replicated in vivo. For example, in vitro studies with *E. faecalis* strain decreased the *C. jejuni* load by two log CFU/g. However, the *E. faecalis* strain could not decrease *C. jejuni*’s load in vivo [86]. In vitro studies with *L. plantarum* N8, N9, ZL5, and *L. casei* ZL4 adhered to the HT-29 cell line and competitively disrupted the adhesion and invasion of *C. jejuni* to the HT-29 cell line [91]. Similarly, *L. paracasei JR*, *L. rhamnosus* 15b, *Y L. lactis,* and *L. lactis FOA* decreased *C. jejuni* adhesion and invasion of the primary chicken cell line [92]. Probiotics enhance the integrity of the intestinal barrier by upregulating the expression of tight junction genes [93]. In a study, the supplementation of *E.coli Nissle* 1917 to the Ht-29 cell line upregulated tight junction genes expression resulting in lower *C. jejuni* intracellular invasion [93].

The supplementation of Poultry Star^®^, (Overland Park, KS, USA) (*E. faecium*, *P. acidilactic*, *L. salivarius*, and *L. reuteri*) via drinking water from the day of hatch to slaughtering decreased *C. jejuni* cecal load by six log CFU/g at 35 days of age [84]. However, studies with Poultry Star^®^ supplementation only decreased *C. jejuni* cecal load by 2 log CFU/g at day 35, and no reduction in *C. jejuni* cecal load at 42 days of age [94]. The variability in probiotic efficacy between the two studies can be due to the difference in experimental design, challenge timing, and cecal microbiota of the broiler birds.

Isolated *B. subtilis* exhibited an anti-*Campylobacter* activity in vitro [95]. The supplementation of *B. subtilis* isolates reduced *C. jejuni* cecal load by one log CFU/g [95]. Motile probiotic bacteria can migrate towards the ceca where *C. jejuni* resides, thus having more chances to eliminate *C. jejuni* [95]. Therefore, the same *B.* subtilis isolates were propagated ten times to increase their motility. The propagated *B. subtilis* supplementation reduced *C. jejuni* cecal load by 2.5 log CFU/g at 21 days of age [95]. Similarly, the supplementation of *B. longum subsp. longum* PCB133 + galactooligosaccharide decreased *C. jejuni* cecal load by 1 log CFU/g at 56 days of age [96]. The oral gavage of *L. salivarius SMXD51*, every 2–3 days from the hatch until day 35, resulted in a 2.5 log CFU/g reduction in *C. jejuni* cecal load [97]. Furthermore, the supplementation of *L. paracasei* J. R, *L. rhamnosus* 15b, *L. lactis Y*, and *L. lactis* FOA combination used seven days before slaughter decreased *C. jejuni* load by five log CFU/g [92]. These data may suggest that multispecies probiotics may be better than single species probiotics in decreasing *C. jejuni* load in the poultry.

Contradictory results were obtained when evaluating the efficacy of probiotics on *C. jejuni* load in broilers. The probiotic bacterial strain, the supplementation dose of probiotics, the route of administration, *C. jejuni* challenge strain, age, sex, and breed of birds used for the study should be considered when evaluating the efficacy of probiotics. Moreover, *C. jejuni* survival in the host depends on the host’s microbiota [98]. Therefore, an interaction between *C. jejuni* and residing microbiota can influence the efficacy of the supplemented probiotic either positively or negatively.

### 5.3. Organic Acids

Organic acids are organic compounds that have acidic properties [99]. Organic acids are differentiated from other acids by having a carboxyl acid -COOH to which hydrogen or an organic compound might be attached [99]. Organic acids can be (1) short-chain fatty acids (SCFAs) (≤C6) such as acetic, lactic, butyric, fumaric, and propanoic acid, (2) medium-chain fatty acids (C7:C10) such as capric, caprylic acid, and (3) long-chain fatty acids (≥C11) such as lauric acid [99]. The gastrointestinal tract of avian species harbors millions of bacteria that produce different metabolites, including organic acids [99]. The antimicrobial mechanism in probiotics is mediated through the production of organic acids [81]. Therefore, supplementing organic acid is expected to impact the bird’s health positively. Organic acids supplementation leads to a decrease in the gut pH, enhancing the proteolytic enzymes and nutrient digestibility [99]. Furthermore, organic acids can act as bacteriostatic and/or bactericidal against gram-negative pathogenic bacteria, making them a suitable antibiotic alternative [83].

The supplementation of 2% formic acid in combination with 0.1% sorbate prevented the colonization of *C. jejuni* [100]. However, supplementing 2% formic acid alone was insufficient to prevent *C. jejuni* colonization [100]. Formic acid lowers the pH of the gut affecting the acid-sensitive bacteria present in the environment. On the other hand, sorbate targets the bacteria by diffusing through the cell membrane and lowering the pH of *C. jejuni*.

In vitro, butyrate supplementation demonstrated a bactericidal effect against *C. jejuni* [101]. However, the supplementation of butyrate-coated micro-beads did not decrease *C. jejuni* colonization [101]. The ineffectiveness of in vivo butyrate supplementation can be attributed to the fast absorption of butyrate by enterocytes. Similarly, the feed acidification with 5.7% lactic acid and 0.7% acetic acid decreased the presence of *C. jejuni* in the feed [102]. However, the limited effect of organic acids on *C. jejuni* colonization in broilers might be attributed to the possibility of being absorbed by the gut microbes before it reaches the ceca [102].

Another study tested different combinations of organic acids, prebiotics, and probiotics against *C. jejuni* infection. Only Adimix^®^ Precision, (Dendermonde, Belgium) (sodium salt butyrate) decreased the cecal load of *C. jejuni* by 2 log CFU/g at 42 days of age [94]. Other compounds containing organic compounds such as lactic acid had a limited ability to decrease *C. jejuni* cecal load. The efficacy of organic acids in controlling enteric pathogens relies heavily on (1) type of SCFAs, (2) dose of SCFAs in feed, (3) buffering capacity of the feed and (4) complex microbiota in the host.

When choosing the type of organic acid for supplementation, the pathogen’s metabolism also should be considered. *C. jejuni* cannot ferment carbohydrates and depends mainly on amino acids and some SCFAs to proliferate in the avian gut [103]. *C. jejuni* utilizes acetate, lactate, fumarate, succinate, and malate as part of its citric acid cycle to satisfy its energy needs [103]. The ability of *C. jejuni* to use SCFAs might explain the ineffectiveness of organic acid supplementation such as lactic acid, formic acid, and acetic acid in the presented studies. Butyrate supplementation might be the organic acid of choice to control *C. jejuni*. Yet, the fact that butyrate is the primary energy source for enterocytes [104] limits its presence in the gut and decreases its ability to fight pathogens.

### 5.4. Bacteriophages

Bacteriophages are viruses ubiquitously found in nature and infect bacterial and archaeal cells [105]. The world contains approximately 10^32^, which is almost 10 times more than the number of bacterial cells on earth [106]. Bacteriophages were discovered in 1915 by Frederick Twort and Félix d’Hérelle in 1917 [107]. Bacteriophages are considered non-pathogenic to humans as they are frequently isolated from human saliva [108] and feces [109]. Humans are frequently exposed to bacteriophages in food and drinking water without any adverse reaction to their consumption [110]. Moreover, bacteriophages dominate the human gut virome [110]. However, the chicken virome is yet to be characterized.

Bacteriophages are suggested as an antibiotic alternative in controlling foodborne pathogens, as they are easy to isolate, have narrow specific, and do not alter the microbiome of the treated host [111]. Though more than 170 *C. jejuni* phages have been documented, the majority of these phages have a narrow spectrum [105] to control foodborne pathogens. *C. jejuni* phages are divided into two categories: lytic and lysogenic. Lytic bacteriophages are preferred as they can lyse the targeted cell immediately [111]. In contrast, the lysogenic bacteriophages are not used as they incorporate into the bacterial genome and transfer virulence factors between bacteria [111].

Lytic *Campylobacter* phages are categorized based on their size into three categories [105]. The first category includes large phages, ranging between 320 and425 kbp, whereas the second category comprises phages that range between 175 and183 and show a high affinity towards *C. jejuni* and *C. coli* [105]. The third category includes *Campylobacter* phages with the smallest size and the greatest affinity and lytic ability toward *C. jejuni* [105]. *Campylobacter* phages are versatile tools that can be incorporated into the preharvest and postharvest control of foodborne pathogens.

The efficacy of two *Campylobacter* phages, CP8 and CP34, was tested for five days post-*C. jejuni* infection, resulting in a 0.5–5 log CFU/g reduction in *C. jejuni* based on the intestinal site and phage dose [112]. The best results in *C. jejuni* load reduction was obtained 24–48 h post bacteriophage supplementation [112]. Similarly, the efficacy was tested with two administration routes (drinking water vs. feed) of a phage cocktail (phiCcoIBB35, phiCcoIBB37, and phiCcoIBB12) [113]. The highest reduction (2 log CFU/g) of *C. jejuni* load was recorded when the cocktail phage was supplemented in the feed [113]. These results indicated the ability of bacteriophages to control *C. jejuni* and highlighted the need to determine the dose and route of administration to achieve the best results.

A comparison between a single phage and a cocktail phage to reduce *C. jejuni* colonization resulted in a maximum of 2.8 log CFU/g cecal *C. jejuni* load in both groups [114]. The single phage resulted in 43% phage resistance, whereas the cocktail phage led to 24% phage resistance [114]. The development of phage resistance limits the using *Campylobacter* phages in controlling *C. jejuni* prevalence. Though the use of phage cocktails might delay the *C. jejuni* development, the effectiveness of *C. jejuni* phages is yet to be determined at the farm scale [114].

### 5.5. Bacteriocins

Bacteriocins are ribosomal synthesized antimicrobial peptides secreted by bacteria. Bacteriocins can act as bacteriostatic and bactericidal against related bacterial species [115]. The secretion of bacteriocins confers the destruction of targeted bacteria without damaging the host. Bacteriocin mode of action is mediated through membrane permeabilization followed by cell lysis [116]. The supplementation of bacteriocins in *C. jejuni* infected broilers efficiently reduces *C. jejuni*’s load and contamination in the food chain [117,118]. Seven-day-old broilers treated with purified encapsulated OR7 bacteriocins produced by *L. acidophilus* NRRL B-30514 significantly reduced *C. jejuni* load [118].

Similarly, supplementing two purified forms, *L. salivarius* NRRL B-30514 and *P. polymyxa* NRRL B-30509, decreased the *C. jejuni* cecal load by three log CFU/g [117].

Recently, reuterin emerged as a promising bacteriocin in controlling *C. jejuni* colonization in broilers. Reuterin is an antimicrobial compound produced during the anaerobic formation of glycerol by *L. reuteri* [119]. Reuterin exhibits a wide antimicrobial spectrum against gram-negative and gram-positive bacteria, yeast, and mold [119]. The mechanism of action of reuterin is mediated through the reaction of acrolein with the thiol groups of glutathione, inhibiting the redox-base defenses and leading to oxidative stress in the targeted bacteria [120]. The genome analysis of *C. jejuni* revealed the absence of glutathione biosynthesis protein, suggesting that *C. jejuni* lacks the ability to detoxify acrolein [121]. The absence of glutathione biosynthesis protein might explain the susceptibility of *C. jejuni* during in vitro studies to reuterin [122].

Bacteriocins production has a high metabolic cost; hence probiotic species will not overproduce it. Supplementing encapsulated bacteriocins with probiotic species might play a prominent role in competitively excluding *C. jejuni* from the avian gut.

### 5.6. Vaccines

Vaccination remains a potentially effective strategy to mitigate the prevalence of foodborne pathogens (*C. jejuni* and *Salmonella*) in poultry production [123]. Vaccination aims to stimulate a mucosal anti-*Campylobacter jejuni* immune response and reduce the *C. jejuni* load at market age. Several vaccine strategies have been developed to control *C. jejuni* in broilers:

#### 5.6.1. Whole Cell Vaccine and Live Attenuated Vaccine

A formalin-killed *C. jejuni* whole cell vaccine containing 2.7 × 10^8^ CFU/mL *C. jejuni* combined with an oil adjuvant or aluminum hydroxide gel adjuvant was inoculated subcutaneously in Japanese Jordi chicken at 37 days of age [124]. The aluminum hydroxide gel adjuvant group received a booster at 58 days of age. The birds were challenged with *C. jejuni* on 72 days of age. Both vaccine groups induced high anti-*C. jejuni* IgG levels [124]. Similarly, a formalin-killed *C. jejuni* whole-cell vaccine was formulated with or without an *E. coli* heat-labile toxin as an adjuvant. The vaccine administration enhanced the anti-*C. jejuni* levels and reduced *C. jejuni* colonization from 16% to 93% in the vaccinated group compared with the non-vaccinated one [125]. However, the *E. coli* heat liable toxin did not increase the immunogenicity of the vaccine [125].

Oxidative stress response plays a significant role in *C. jejuni*’s enteric lifestyle. *C. jejuni* oxidative stress defense mutant shows a low ability to persist in the avian gut. In this study, birds were orally gavaged with 0.5 mL of *C. jejuni* ΔahpC mutant at 3 and 7 days of age, followed by a challenge (WT *C. jejuni*) at 14 days of age [126]. The pre-colonization of broilers with the *C. jejuni* ΔahpC mutant decreased the *C. jejuni* by three log CFU/g reduction at 42 days of age [126]. These results suggest that the *C. jejuni* ΔahpC mutant has the potential to be used at the farm level to control *C. jejuni* at the preharvest stage; however, more safety studies are required at a farm level.

#### 5.6.2. Crude Cell Lysate

The efficacy of a nanoparticle vaccine composed of poly lactide-co-glycolide nanoparticle (NP) and encapsulated 25, 125, or 250 μg outer membrane of *C. jejuni* was evaluated against *C. jejuni* [127]. The subcutaneous route induced the highest immune response in vaccinated broilers and decreased the *C. jejuni* load by 5.7 logCFU/g in the ceca [127]. Similarly, the oral delivery of *C. jejuni* oral lysate reduced the *C. jejuni* load in layer and broiler chickens by 2.24 log CFU/g and 2.14 log CFU/g, respectively, at 22 days post-infection [128].

#### 5.6.3. Subunit Vaccine

Type VI secretion system (T6SS) enables bacteria to infect neighboring cells and plays a vital role in inter-bacterial competition and bacterial communication with the host’s cells [129]. In *C. jejuni*, the type VI secretion system (T6SS) plays a role in evading the immune system and bacterial survival [130]. A subunit vaccine was formulated from a purified 50 µg of recombinant hemolysin co-regulated protein (RHCP) entrapped in chitosan sodium tripolyphosphate nanoparticles [131]. The broilers were orally gavaged with subunit vaccine at 7 days of age and then boosted at 14 and 21 days of age [131]. The vaccinated broilers were then challenged with *C. jejuni* at 28 days of age. The vaccinated group had one log CFU/g reduction of *C. jejuni* load in the ceca [131].

#### 5.6.4. Bacterial Vector-Based Vaccine

Live or genetically engineered bacterial strains emerge as potential vaccine candidates against enteric pathogens. Bacteria that are avirulent to chickens and elicit an immune response are considered suitable vectors. These vectors can present *C. jejuni* virulent antigens to the birds’ immune system. *C. jejuni* mutants show a transient colonization pattern in the chicken gut and do not persist enough to activate an immune response [132]. Avirulent *Salmonella* and *Lactobacillus* strains are the bacterial vectors of choice for creating a bacterial vector-based vaccine against *C. jejuni*. An avirulent *Salmonella Typhimurium* χ3987 strain expressing CjaA was orally gavaged in broilers at 1 and 14 days of age (booster) [133]. At 28 days of age, the broiler was challenged by *C. jejuni*. The vaccine inoculation led to 6.0 log CFU/g reductions in the *C. jejuni* cecal load [133].

Finally, phase variation and strain differences in *C. jejuni*, *C. coli*, and *C. lari* complicate the development of a potential vaccine that can decrease campylobacteriosis around the globe.

### 5.7. Quorum Sensing Inhibitors

In broilers, *C. jejuni* cecal load can reach up to 1 × 10^9^ CFU/g [134]. *C. jejuni* can detect and respond to rapid changes in bacterial densities using quorum sensing [135]. Quorum sensing is a cell-to-cell communication in which bacteria produce, detect, and respond to signaling molecules known as autoinducers [136]. The accumulation of autoinducers happens in a density-dependent manner [135]. When the autoinducer concentration reaches a certain threshold, it leads to the activation of a signal cascade [137]. The signal cascade alters gene expression, resulting in morphological changes in the bacterium that aids its survival in the environment [137].

At first, quorum sensing studies in *C. jejuni* identified a gene that encodes an orthologue of the LuxS system that mediates the production of autoinducer-2 (AI-2) [138]. In the same study, the *C. jejuni* luxS mutants showed a decreased motility in semisolid media, indicating a key role of luxS in regulating darting motility in *C. jejuni* [138]. Furthermore, the role of luxS in host colonization was evaluated in a study testing the colonization ability of the *C. jejuni* luxS mutant strain vs. the *C. jejuni* wild-type strain [139]. The luxS mutant showed decreased colonization capacity in chicken seven days post-inoculation; however, some birds inoculated with the luxS mutant strains maintained a similar level of colonization compared with groups inoculated with the wild-type strain [139]. Furthermore, a competitive fitness experiment between the wild-type and luxS mutant showed a decrease in the recovery of the mutant in comparison with the wild-type, indicating an important role of luxS in *C. jejuni* fitness [140].

The imminent role of luxS in *C. jejuni*’s adaption to environmental conditions [141], expression of virulence factors [140], and biofilm formation [142] make it a potential target for controlling *C. jejuni* infection. In vitro, (-)-α-pinene showed an anti-quorum sensing activity against *C. jejuni* by decreasing the *C. jejuni* quorum signaling by more than 80% [143]. The supplementation of 250 mg/L of (-)-α-pinene in *C. jejuni*-challenged broilers resulted in 0.8 log CFU/g reduction in the cecal load [143]. (-)-α-pinene inhibitory activity against *C. jejuni* is attributed to (-)-α-pinene ability to inhibit efflux pump activity and quorum sensing, which play a crucial role in colonizing the host [143]. Thus, (-)-α-pinene can potentially contribute to the control of *C. jejuni* in broilers.

Citrus extracts decreased motility and biofilm formation in *E. coli* O157:H7, *S. typhimurium*, and *P. aeruginosa*. Similarly, citrus extracts inhibited *C. jejuni* autoinducer-2 quorum sensing, resulting in lower motility and lower biofilm formation [144]. Similarly, *Sedum rosea* (roseroot) extract decreased *C. jejuni* quorum signaling by more than 90% and decreased *C. jejuni* invasion of INT407 cells by 80% [145]. These results demonstrate the ability of natural phenolic compounds to alter the quorum sensing in *C. jejuni*, resulting in a lower fitness in *C. jejuni*. Quorum inhibitory compounds are a promising tool to control *C. jejuni* in poultry. However, additional studies are needed to determine the required dose and treatment period to decrease the load of *C. jejuni* in poultry. The control strategies of *C. jejuni* are summarized in Figure 3.

## 6. Conclusions

Chicken around the globe remains the main reservoir for campylobacteriosis in humans. With the increase in campylobacteriosis worldwide, antibiotic resistance in *C. jejuni* and increased post-*C. jejuni* infection complications (such as GBS and MRS), looking for a suitable control strategy becomes the need of the hour. A multi-hurdle approach is needed to ensure the control of foodborne pathogens from farm to fork. Strict biosecurity combined with feed additives and a suitable vaccine (if developed) might be the method of choice to control *C. jejuni* in broiler production.

## Figures and Tables

**Figure 1 microorganisms-10-02134-f001:**
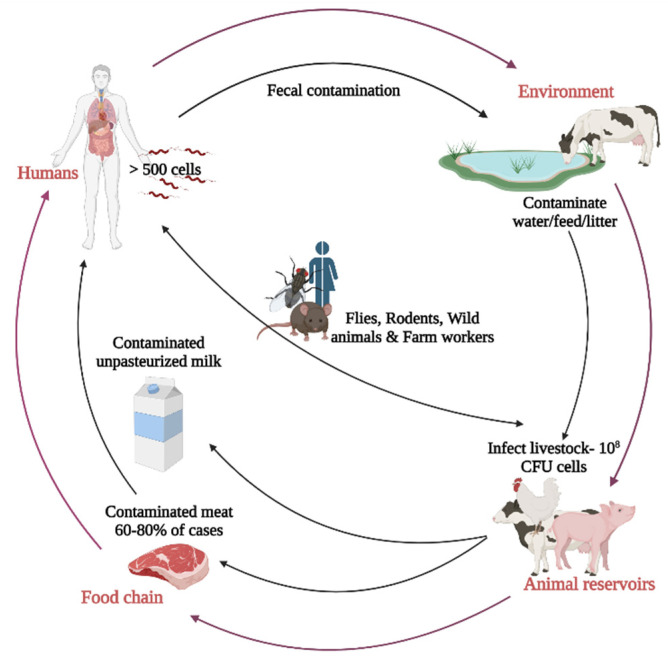
Reservoirs and routes of transmission of *C. jejuni*. Created with Biorender.com (accessed on 20 October 2022).

**Figure 2 microorganisms-10-02134-f002:**
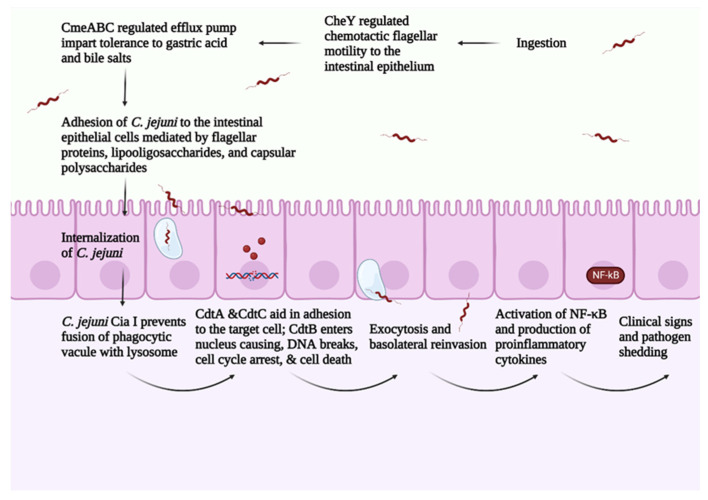
Overview of *C. jejuni* pathogenesis. Created with Biorender.com (accessed on 20 September 2022).

**Figure 3 microorganisms-10-02134-f003:**
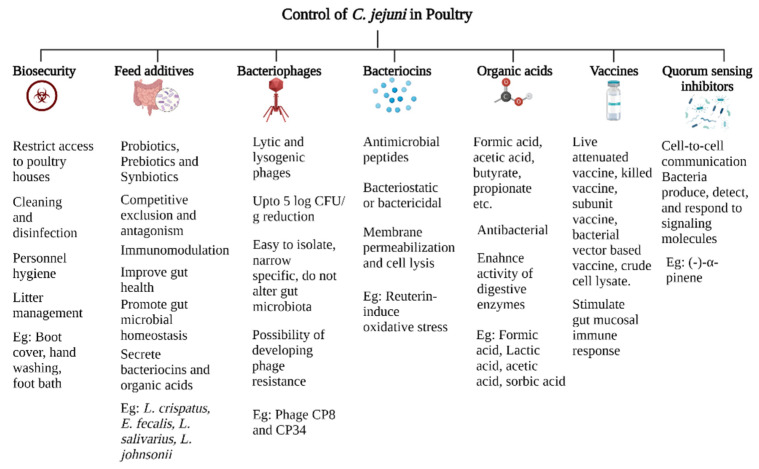
Control strategies of *C. jejuni* in poultry. Created with Biorender.com (accessed on 20 October 2022).

## Data Availability

No new data were created or analyzed in this study. Data sharing is not applicable to this article.

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
