# Peer review of "Campylobacter jejuni in Poultry: Pathogenesis and Control Strategies"

_microorganisms, 2022, doi:10.3390/microorganisms10112134_

Round 1
Reviewer 1 Report
Hakeem et al. assembled a significant amount of information dealing with Campylobacter jejuni in poultry. The authors outline the tremendous problems this pathogen causes in the human food chain and discuss various strategies that are currently investigated to reduce its prevalence. Many approaches are still in the early research stages, no convincing strategy to fight Campylobacter in poultry has as yet been established in the field. Nevertheless, the review achieves its goals in giving a comprehensive overview of the current efforts.
The review would greatly profit from some skilled language editing, but otherwise requires only minor revisions.
Specific comments:
In the entire manuscript C. jejuni is lacking a “space” between the abbreviation dot and the second word. Also all other species names in latin should be written like that.
Introduction, lines 55-56: … synbiotic, bacteriocins, bacteriophages, vaccines, and organic… comma after bacteriophages is missing.
Lines 80-82: The second part of the sentence “Different stress conditions lead to morphological changes in C.jejuni, oxygen-rich compounds that change from spiral shape to coccoid form [22]” is incomplete and does not make sense in this form. As a suggestion: “ …, for example, oxygen-rich compounds can lead to a change from spiral shape to a coccoid form.”
Line 108: … hindered by high O2 and… please subscript the 2.
Lines 172-173: The sentence “Furthermore, the flagella include the type 3 secretion system, responsible for delivering effector proteins needed for cellular invasion [20]” does not make any sense. The flagella cannot include a type 3 secretion system! Could it be that the genes for the type 3 secretion system are located in the flagella operon? The sentence should be rewritten to clarify the point of the authors.
Line 196 and elsewhere in the text: . “anti-campylobacter jejuni properties” [68]. Please capitalize Campylobacter and write the latin species name in italics.
Line 96: Please specify if you are talking about laying hens or turkey here: “… C.jejuni cecal load by 1 log CFU/g at 56 days of age [96].” Broilers rarely reach day 56.
Line 324: “… such as Capric, Caprylic acid, and 3) long-chain fatty acids” arbitrary capitalization should be removed.
Line 367: The name “Felix d’Herrlle” is misspelled. Correct : Félix d’Hérelle
Line 376: phages not “C. jejuni phases”
Line 420: please capitalize the latin species names, i.e. Salmonella
Line 430: as above mentioned, E. coli
Line 464: “… and can elect an …”? elicit?
Line 468: as above, Salmonella and Lactobacillus
Author Response
Hakeem et al. assembled a significant amount of information dealing with Campylobacter jejuni in poultry. The authors outline the tremendous problems this pathogen causes in the human food chain and discuss various strategies that are currently investigated to reduce its prevalence. Many approaches are still in the early research stages, no convincing strategy to fight Campylobacter in poultry has as yet been established in the field. Nevertheless, the review achieves its goals in giving a comprehensive overview of the current efforts.
The review would greatly profit from some skilled language editing, but otherwise requires only minor revisions.
Specific comments:
- In the entire manuscript C. jejuni is lacking a “space” between the abbreviation dot and the second word. Also all other species names in latin should be written like that.
Addressed. The Latin name of C. jejuni and other bacteria were fixed in the manuscript.
- Introduction, lines 55-56: … synbiotic, bacteriocins, bacteriophages, vaccines, and organic… comma after bacteriophages is missing.
Addressed
- Lines 80-82: The second part of the sentence “Different stress conditions lead to morphological changes in C.jejuni, oxygen-rich compounds that change from spiral shape to coccoid form [22]” is incomplete and does not make sense in this form. As a suggestion: “ …, for example, oxygen-rich compounds can lead to a change from spiral shape to a coccoid form.”
Addressed
- Line 108: … hindered by high O2 and… please subscript the 2.
Addressed
- Lines 172-173: The sentence “Furthermore, the flagella include the type 3 secretion system, responsible for delivering effector proteins needed for cellular invasion [20]” does not make any sense. The flagella cannot include a type 3 secretion system! Could it be that the genes for the type 3 secretion system are located in the flagella operon? The sentence should be rewritten to clarify the point of the authors.
The type 3 secretion system in the Campylobacter species is located within the flagella. The following reference addresses the presence of a flagellar type 3 secretion system in C.jejuni.
Christensen JE, Pacheco SA, Konkel ME. Identification of a Campylobacter jejuni-secreted protein required for maximal invasion of host cells. Mol Microbiol. 2009 Aug;73(4):650-62. doi: 10.1111/j.1365-2958.2009.06797.x. Epub 2009 Jul 14. PMID: 19627497; PMCID: PMC2764114.
- Line 196 and elsewhere in the text: . “anti-campylobacter jejuni properties” [68]. Please capitalize Campylobacter and write the latin species name in italics.
Addressed
- Line 96: Please specify if you are talking about laying hens or turkey here: “… C.jejuni cecal load by 1 log CFU/g at 56 days of age [96].” Broilers rarely reach day 56.
The authors of this experiment used a 40-day-old broiler ( Kabir strain). And the reduction of C. jejuni was recorded at 56 days of age.
Reference: Loredana Baffoni, Francesca Gaggìa, Diana Di Gioia, Cecilia Santini, Luca Mogna, Bruno Biavati, A Bifidobacterium-based synbiotic product to reduce the transmission of C. jejuni along the poultry food chain,International Journal of Food Microbiology,Volume 157, Issue 2,2012,Pages 156-161,ISSN 0168-1605, https://doi.org/10.1016/j.ijfoodmicro.2012.04.024
- Line 324: “… such as Capric, Caprylic acid, and 3) long-chain fatty acids” arbitrary capitalization should be removed.
Addressed
- Line 367: The name “Felix d’Herrlle” is misspelled. Correct : Félix d’Hérelle
Addressed
- Line 376: phages not “C. jejuni phases”
Addressed
- Line 420: please capitalize the latin species names, i.e. Salmonella
Addressed
- Line 430: as above mentioned, E. coli
Addressed
- Line 464: “… and can elect an …”? elicit?
Addressed
- Line 468: as above, Salmonella and Lactobacillus
Addressed
Reviewer 2 Report
The review article by Hakeem et al., entitled “Campylobacter jejuni in Poultry: Pathogenesis and Control Strategies” is good to study but it lacks novelty.
1. The objectives are very concise, i.e., morphology, pathogenesis etc which are already well-known in the literature.
2. Figures are also very basic; no new approaches are represented by any of the figures.
3. Authors must extend section 5.5 bacteriocins.
4. Authors have mainly focussed on the known information about the pathogen and its pathogenesis. Authors must try to add the latest advancement and prospects.
5. The manuscript required major revision.
Author Response
The review article by Hakeem et al., entitled “Campylobacter jejuni in Poultry: Pathogenesis and Control Strategies” is good to study but it lacks novelty.
- The objectives are very concise, i.e., morphology, pathogenesis etc which are already well-known in the literature.
Addressed in the manuscript
- Figures are also very basic; no new approaches are represented by any of the figures.
Addressed in the new figures.
- Authors must extend section 5.5 bacteriocins.
Addressed in section 5.5. Bacteriocins.
- Authors have mainly focussed on the known information about the pathogen and its pathogenesis. Authors must try to add the latest advancement and prospects.
Addressed in section 5.7
- The manuscript required major revision.
Adderesed
Round 2
Reviewer 2 Report
Manuscript can be accepted in the current form.